# pH Dependence of *T*_2_ for Hyperpolarizable ^13^C-Labelled Small Molecules Enables Spatially Resolved pH Measurement by Magnetic Resonance Imaging

**DOI:** 10.3390/ph14040327

**Published:** 2021-04-02

**Authors:** Martin Grashei, Christian Hundshammer, Frits H. A. van Heijster, Geoffrey J. Topping, Franz Schilling

**Affiliations:** Department of Nuclear Medicine, TUM School of Medicine, Klinikum rechts der Isar, Technical University of Munich, D-81675 Munich, Germany; martin.grashei@tum.de (M.G.); christian.hundshammer@wacker.com (C.H.); frits.van.heijster@tum.de (F.H.A.v.H.); geoff.topping@tum.de (G.J.T.)

**Keywords:** *T*_2_ relaxation time constant, pH, [1-^13^C]pyruvate, [1-^13^C]acetate, ^13^C-labeled biomolecules, hyperpolarization, hyperpolarized MRI, magnetic resonance spectroscopy

## Abstract

Hyperpolarized ^13^C magnetic resonance imaging often uses spin-echo-based pulse sequences that are sensitive to the transverse relaxation time *T*_2_. In this context, local *T*_2_-changes might introduce a quantification bias to imaging biomarkers. Here, we investigated the pH dependence of the apparent transverse relaxation time constant (denoted here as *T*_2_) of six ^13^C-labelled molecules. We obtained minimum and maximum *T*_2_ values within pH 1–13 at 14.1 T: [1-^13^C]acetate (*T*_2,min_ = 2.1 s; *T*_2,max_ = 27.7 s), [1-^13^C]alanine (*T*_2,min_ = 0.6 s; *T*_2,max_ = 10.6 s), [1,4-^13^C_2_]fumarate (*T*_2,min_ = 3.0 s; *T*_2,max_ = 18.9 s), [1-^13^C]lactate (*T*_2,min_ = 0.7 s; *T*_2,max_ = 12.6 s), [1-^13^C]pyruvate (*T*_2,min_ = 0.1 s; *T*_2,max_ = 18.7 s) and ^13^C-urea (*T*_2,min_ = 0.1 s; *T*_2,max_ = 0.1 s). At 7 T, *T*_2_-variation in the physiological pH range (pH 6.8–7.8) was highest for [1-^13^C]pyruvate (Δ*T*_2_ = 0.95 s/0.1pH) and [1-^13^C]acetate (Δ*T*_2_ = 0.44 s/0.1pH). Concentration, salt concentration, and temperature alterations caused *T*_2_ variations of up to 45.4% for [1-^13^C]acetate and 23.6% for [1-^13^C]pyruvate. For [1-^13^C]acetate, spatially resolved pH measurements using *T*_2_-mapping were demonstrated with 1.6 pH units accuracy in vitro. A strong proton exchange-based pH dependence of *T*_2_ suggests that pH alterations potentially influence signal strength for hyperpolarized ^13^C-acquisitions.

## 1. Introduction

Since the introduction of dissolution dynamic nuclear polarization in 2003 [1], several hyperpolarized ^13^C-labelled biomolecules have been applied in preclinical [2] and clinical studies [3], establishing new ways to generate magnetic resonance imaging (MRI) contrast. The metabolic conversion of [1-^13^C]pyruvate to its downstream metabolites [1-^13^C]lactate, [1-^13^C]alanine and ^13^C-bicarbonate has been investigated as an imaging biomarker for the detection and evaluation of various pathologies such as cancer [4], inflammation [5], and related changes in pH [6], as well as the functionality of various organs, such as the heart [7], liver [8], or kidney [9], and of perfusion [10]. Metabolically inert compounds such as ^13^C-urea and [^13^C,^15^N_2_]urea have also been introduced, enabling the assessment of perfusion and kidney function from signal intensity maps [9]. Hyperpolarized ^13^C-labelled molecules can also be used as sensors of physicochemical properties by exploiting changes in longitudinal and transverse relaxation times or chemical shift, e.g., for the detection of metal ion concentrations [11] or pH [6,12].

To sense these properties, the initially created hyperpolarized longitudinal magnetization must be excited in MRI experiments*,* thereby converting the longitudinal magnetization into transverse magnetization. This transverse magnetization then precesses about the static magnetic field axis while decaying exponentially with a time constant *T*_2_, known as the transverse relaxation time constant. This decay mainly occurs because the individual spins, composing the net transverse magnetization, experience microscopic field fluctuations over time. These fluctuations induce small changes in the spins’ Larmor frequencies, thereby causing the spins to lose synchrony over time, i.e., to dephase, decreasing the measurable net signal.

*T*_2_ limits preparation and acquisition time and thereby influences the achievable spectral and spatial resolution and signal strength; therefore, a long *T*_2_ is desirable for hyperpolarized ^13^C-MRI experiments. Nevertheless, *T*_2_ weighting or direct measurement of *T*_2_ can also be exploited to generate contrast in hyperpolarized ^13^C-MRI, whereby dependencies of *T*_2_ on biomarkers such as pH might be of interest for in vivo applications.

However, *T*_2_ measurements involve acquisitions at multiple echo times that lead to additional dephasing beyond *T*_2_. This includes spins diffusing between refocusing pulses [13] and imperfect refocusing [14], potentially leading to a mixture of *T*_1_ and *T*_2_ decay. The measured *T*_2_ is thus an apparent transverse relaxation time constant (Appendix B). Despite this discrepancy, the term *T*_2_ is used in the following when reporting results, while explicitly stating here that apparent *T*_2_ relaxation times are meant.

Changes in the apparent transverse relaxation time, as demonstrated by *T*_2_-mapping for hyperpolarized ^13^C-urea [15], have been shown to detect alterations of tissue oxygenation [16], protein content [16], viscosity [17] or restricted diffusion due to cellular uptake [17]. Furthermore, *T*_2_ measurements in hepatocellular carcinoma showed differences in *T*_2_ between healthy and tumor tissue for [1-^13^C]alanine and [1-^13^C]lactate [18], with further heterogeneity in *T*_2_ indicated by the necessity of multi-exponential fitting of the transverse magnetization decay curves of healthy [15,17,19] and tumor tissue [19,20].

Many approaches for the assessment of metabolism using hyperpolarized magnetic resonance spectroscopy and imaging rely on the acquisition of spin echoes and are inherently sensitive to *T_2_*, such as point-resolved spectroscopy (PRESS) [18,21], fast spin echo [16,22,23], double spin echo [8,24], multi-echo spin echo [20], or balanced steady-state free precession (bSSFP) sequences [9,15,17,25,26]. However, to the best of our knowledge, the sensitivity of these sequences to alterations of *T*_2_ and heterogeneity in *T*_2_ within the imaged object has so far not been considered for the calculation of metabolic conversion rates. This is especially important for sequences that employ echo times on the time scale of the *T*_2_ relaxation constants of the metabolites being imaged.

Consequently, the *T*_2_ relaxation time constants of several commonly imaged biomolecules, such as [1-^13^C]pyruvate [19,20,25,27,28], [1-^13^C]lactate [18,19,20,25,27], [1-^13^C]alanine [18,19], ^13^C-urea [15], [^13^C,^15^N_2_]urea [15,16,17], ^13^C-bicarbonate [19,26], bis-1,1-(hydroxymethyl)-1-^13^C-cyclopropane-^2^H_8_ (HP001) [27] or [1-^13^C]acetate [26] have been measured at various magnetic field strengths; in vivo for various organs or diseases, or in vitro for various viscosities or media. In addition to those influencing factors, further parameters that are altered in vivo, such as tissue [29] or blood [30,31,32] buffer capacity and pH [33,34], might affect *T_2_* but have not yet been explored.

In this work, the pH dependence of the *T*_2_ relaxation time constant of several ^13^C-labelled biomolecules that are commonly used for hyperpolarized magnetic resonance imaging is investigated. For the two compounds with the strongest pH dependence of *T*_2_ in the physiological pH range, namely, [1-^13^C]pyruvate and [1-^13^C]acetate, further factors influencing *T*_2_ are examined, including concentration, temperature, buffer capacity, and salt concentration. Furthermore, *T*_2_ mapping is used to generate images with strong pH-based contrast in buffer-free aqueous solutions using [1-^13^C]acetate as a pH sensor.

## 2. Results

The apparent *T*_2_ relaxation time constants of some of the most commonly applied hyperpolarized ^13^C-labelled small molecules, namely [1-^13^C]acetate, [1-^13^C]pyruvate, [1-^13^C]lactate, [1-^13^C]alanine and ^13^C-urea, were measured across the pH range 1–13. For [1,4-^13^C_2_]fumarate, the *T*_2_ relaxation time constant was measured across the pH range 4–13, because its solubility strongly decreases for pH < 4.

### 2.1. pH Dependency of T_2_ for Commonly Used Hyperpolarized ^13^C-Labelled Small Molecules

In the strongly acidic (*T*_2_ = 23.1 s, pH 1.02) and basic (*T*_2_ = 25.8 s, pH 12.97) regimes (Figure 1a), [1-^13^C]acetate showed ^13^C-*T*_2_ relaxation time constants in the order of its *T*_1_ at 14.1 T (*T*_1_ = 27 s, pH 1; *T*_1_ = 39 s, pH 13; [35]), with *T*_2_ values in the basic regime being slightly larger compared to the acidic regime. Furthermore, *T*_2_ exhibited a linear decrease of one order of magnitude (*T*_2_ = 2.1 s, pH 4.52) towards its p*K*_a_ = 4.76. This slope was flattened towards neutral to slightly basic pH values, because water has a value p*K*_a_ = 6.81 at 37 °C in this pH regime [36].

[1-^13^C]pyruvate (Figure 1b) had its maximum *T*_2_ at pH 8.81 (*T*_2_ = 18.7 s, pH 8.81). Towards even stronger alkaline pH values, *T*_2_ decreased by more than two orders of magnitude (*T*_2_ = 0.1 s, pH 12.04). Addition of TRIS buffer (p*K*_a_ = 8.07) also decreased the *T*_2_ of pyruvate by 37% (pH 8.07–8.20), predominantly in pH regimes close to the p*K*_a_ value of the buffer. At even more acidic pH values, close to pyruvate’s p*K*_a_ = 2.50, *T*_2_ was further shortened (1.3 s, pH 2.03).

In contrast, [1-^13^C]lactate showed a global *T*_2_ maximum at strongly acidic pH values (Figure 1c, *T*_2_ = 12.6 s, pH 1.03). Towards less acidic pH values around its p*K*_a_ = 3.86, *T*_2_ strongly decreased and had a minimum at neutral pH (*T*_2_ = 0.7 s, pH 6.90). Towards more alkaline pH values, *T*_2_ of [1-^13^C]lactate increased, reaching a maximum *T*_2_ around pH 11 (*T*_2_ = 7.6 s, pH 10.72). Beyond this pH, more alkaline milieus led to a reduction in *T*_2_ (*T*_2_ = 5.8 s, pH 12.95).

[1-^13^C]alanine showed an overall decrease in *T*_2_ from its maximum at basic pH milieus (Figure 1d, *T*_2_ = 10.6 s, pH 12.45) towards acidic pH milieus, with additional reductions in *T*_2_ at the p*K*_a_ values of the amide (p*K*_a1_ = 9.87) and the carboxyl group (p*K*_a2_ = 2.35). In addition, *T*_2_ decreased by one order of magnitude from basic to acidic pH regimes (*T*_2_ = 0.8 s, pH 1.61).

In strong contrast, ^13^C-urea, which does not exchange protons with its aqueous solvent environment, showed an almost constant value for *T*_2_ across the entire pH range (Figure 1e, ${\bar{T}}_{2}$ = 0.1 s), with a relatively low absolute *T*_2_ value compared to the other investigated ^13^C-labelled molecules (Figure 1a–f).

The *T*_2_ of [1,4-^13^C_2_]fumarate was also shortened for pH values (*T*_2_ = 4.4 s, pH 4.04) near its carboxyl group’s p*K*_a_ values (p*K*_a1_ = 3.03, p*K*_a2_ = 4.44), as well as around pH 7. For alkaline pH values above pH 8, *T*_2_ reached its maximum values (*T*_2_ = 13.5–17.0 s, pH 12.97–13.02).

[1-^13^C]acetate in the basic regime exhibited the highest overall measured *T*_2_ (*T*_2,max_ = 27.7 s, pH 10.90) with a 13.18-fold increase (ratio *T*_2,max_/*T*_2,min_) compared to its measured minimum *T*_2_ (*T*_2,min_ = 2.1 s, pH 4.52). Especially notable in the context of applications in hyperpolarized ^13^C-MRI, [1-^13^C]acetate’s *T*_2_ value at physiological pH 7.4 (*T*_2_ = 14.7 s, pH 7.4) was also the longest of all molecules investigated in this work. Furthermore, *T*_2_ of [1-^13^C]acetate also exhibited a strong sensitivity to alterations in the physiological pH range 6.8–7.8 [37,38] (mean Δ*T*_2_ = 0.44 s/0.1 pH).

Maximum and minimum *T*_2_ values, *T*_2_ values at physiological pH, ratio between maximum and minimum *T*_2_ values and pH sensitivity of *T*_2_ in the physiological pH range are listed for all investigated molecules in Table 1.

As pointed out previously, ^13^C-urea showed low to no sensitivity of *T*_2_ to pH variations in the physiological range (mean Δ*T*_2_ < 0.001 s/0.1pH) and little difference between the maximum and minimum measured *T*_2_ (ratio *T*_2,max_/*T*_2,min_ = 1.2). In contrast, [1-^13^C]pyruvate exhibited a more than 200-fold change between its minimum and maximum *T*_2_ across the entire pH range. Furthermore, [1-^13^C]pyruvate showed the largest sensitivity of *T*_2_ to pH in the physiological pH range of all molecules (mean Δ*T*_2_ = 0.95 s/0.1pH). [1-^13^C]lactate showed a decrease (pH 6.8–6.9) as well as an increase in relaxation time constant values with increasing pH from pH 6.9 to pH 7.8, with a minimum close to pH 6.9.

### 2.2. pH, Temperature and Salt Concentration Dependence of T_2_ for [1-^13^C]acetate and [1-^13^C]pyruvate

Figure 1a,b and Table 1 indicate that [1-^13^C]acetate and [1-^13^C]pyruvate showed a strong linear pH dependence of *T*_2_ in the physiological pH range (pH 6.8–7.8) [37,38,39]. Therefore, both molecules were considered as potential candidates for in vivo pH imaging via *T*_2_ mapping with hyperpolarization techniques for signal enhancement.

The overall pH–*T*_2_ dependence of [1-^13^C]acetate at 7 T (Figure 2a) was similar to higher-field measurements at 14.1 T (Figure 1a), but exhibited higher absolute *T*_2_ values (*T*_2_ = 40.0 s, pH 11.84–12.68). However, at neutral to slightly basic pH, there was a weaker sensitivity to pH changes at 7 T compared to higher field measurements. [1-^13^C]pyruvate showed pH-dependent *T*_2_ relaxation time constants at 7 T (Figure 2b) of similar magnitude (*T*_2,max_ = 23.4 s, pH 8.12) compared to 14.1 T (Figure 1b), however also exhibited a local maximum (*T*_2_ = 8.8 s, pH 3.98) at slightly acidic pH and a local minimum at pH 6.00 (*T*_2_ = 5.3 s, pH 5.87).

The dependence of *T*_2_ on temperature, salt concentration, concentration and magnetic field strength was also investigated. For [1-^13^C]acetate, the temperature dependence of *T*_2_ for three different pH values in the range pH = 8.81–9.43 was measured (Figure 3a). Offsets between the different curves are attributed to the different solution pH values for each curve. Only a moderate increase across a wide temperature range (<50% maximum across 30 °C span) was observed. Addition of NaCl salt (Figure 3b) showed a moderate decrease (<25%) in *T*_2_ up to 1 M salt concentration for two titrations at pH = 8.23 or 9.32. For unbuffered [1-^13^C]pyruvate near pH 1.5, *T*_2_ showed an exponential reduction upon increases in temperature from 15 to 50 °C (Figure 3c). Addition of NaCl salt showed similar effects (Figure 3d) as for [1-^13^C]acetate, with the *T*_2_ relaxation time constant only being reduced by 16% upon the addition of 1 M NaCl. Furthermore, both molecules showed a strong increase in *T*_2_ with decreasing concentration (Appendix A); dilution from 250 mM to 50 mM led to an approximately 100% increase in *T*_2_, while increasing the concentration from 250 mM to 1000 mM resulted in a decreased *T*_2_ of [1-^13^C]acetate and [1-^13^C]pyruvate by up to 50% and less than 24%, respectively.

In addition, for [1-^13^C]pyruvate, the influence of magnetic field strength on *T*_2_ was assessed by comparing measurements at 1 T, 7 T and 14.1 T, which showed moderate decreases in *T*_2_ with higher magnetic field, particularly at slightly acidic and basic pH regimes (Appendix A).

To assess the relevance of concentration, salt concentration, temperature, and pH for *T*_2_ measurements in vivo, a realistic range of parameter values centered around a physiological reference value was selected. *T*_2_ values for the reference values as well as for the upper and lower bounds of these parameters were derived from the curves in Figure 3a–d and Appendix A. The results are listed for [1-^13^C]acetate in Table 2 and for [1-^13^C]pyruvate in Table 3.

The ^13^C-labelled compounds are typically injected at a concentration of 60–100 mM with an injection volume to blood volume ratio of 1:10 [12], thereby potentially resulting in in vivo variations between 5 mM to 80 mM, with values near 10 mM being likely blood concentrations. Variations of the concentration within this range resulted in *T*_2_ changes of up to 45.4% for [1-^13^C]acetate and 23.6% for [1-^13^C]pyruvate. Whole blood typically contains approximately 150 mM dissolved salt, with values below 120 mM and above 180 mM indicating severe hypo- and hypernatremia, respectively [40]. Such variations account for up to 8.5% and 2.0% change in *T*_2_ for [1-^13^C]acetate and for [1-^13^C]pyruvate, respectively. In vivo temperature alterations during magnetic resonance (MR) acquisitions may plausibly cover regions between 35 °C and 39 °C, with 37 °C being normal human body temperature. For [1-^13^C]acetate, this amounted to only a 1.6% change in *T*_2_, while [1-^13^C]pyruvate showed up to 10.6% deviation in *T*_2_ within this temperature range.

Finally, the effect of pH on *T*_2_ within a potential in vivo blood pH range from 6.8 [37] to 7.8 [38], with pH 7.4 being physiological blood pH, was evaluated. This revealed a strong alteration of *T*_2_ by up to 16.9% for [1-^13^C]acetate and up to 41.2% for [1-^13^C]pyruvate within these boundaries, relative to the physiological reference pH. Thus, concentration and pH were shown to have the strongest effect on the transverse relaxation time constants for these molecules under physiologically plausible conditions.

### 2.3. pH Imaging in Aqueous Solutions Using [1-^13^C]acetate

pH strongly influenced *T*_2_ at physiological pH values for [1-^13^C]acetate. Therefore, the ability to spatially resolve pH values using *T*_2_ mapping was investigated. Three tubes of [1-^13^C]acetate containing solutions with three different pH values near physiological pH (6.14, 7.10, 9.21) showed strongly differing *T*_2_ values when measured for each tube separately (Figure 4a, red). Signal time curves (Figure 4b) from the acquired echo images and averaged across each tube showed long-lasting signals, with *T*_2_ values almost identical to those measured with a Carr-Purcell-Meiboom-Gill (CPMG) train without spatial encoding gradients (Figure 4a, black). *T*_2_ maps were generated by voxel-wise fitting (Appendix A) to the echo image intensities. These maps show good homogeneity of the calculated *T*_2_ values across each tube region of interest (ROI) (Figure 4c) and have a high pH contrast between tubes (green encircled ROI: mean *T*_2_ = 6.7 ± 0.2 s; red encircled ROI: mean *T*_2_ = 5.6 ± 0.1 s; magenta encircled ROI: mean *T*_2_ = 12.4 ± 0.2 s). Slight differences in the absolute *T*_2_ values between CPMG and rapid acquisition relaxation enhancement (RARE) values are likely caused by additional diffusion-weighting from the applied imaging gradients leading to a shortening of the apparent transverse relaxation times. The small differences between imaging and non-imaging fit results indicate that diffusion related signal loss in the imaging sequence could be minimized sufficiently using short echo times (Appendix B) and low encoding gradient strengths to reproduce the relaxation time constants measured by the non-imaging CPMG methods.

By fitting the titration curve in Figure 2a with a linear relationship in the range pH 6.14–9.21 (15 data points), calibration curves between pH and *T_2_* can be determined, allowing pH maps to be calculated from *T_2_* maps (Figure 4d). The derived *T*_2_ values from the CPMG measurements (Figure 4a) cannot be perfectly represented with a linear relationship; therefore, pH maps show systematic uncertainty of up to 1.6 pH units (green encircled ROI: mean pH = 5.87 ± 0.06, true pH = 7.10; red encircled ROI: mean pH = 5.56 ± 0.01, true pH = 6.14; magenta encircled ROI: mean pH = 7.62 ± 0.07, true pH = 9.21). However, tubes of different pH remain well-distinguishable, with a calculated pH heterogeneity of less than 0.1 pH unit within each tube. To further evaluate the ability to translate this pH mapping method for in vivo applications, *T*_2_ of hyperpolarized [1-^13^C]pyruvate was measured at 25 mM at 1 T and thermal [1-^13^C]pyruvate was measured at 600 mM at 7 T in human blood (Appendix C). Despite some indication for larger *T*_2_ values at more acidic blood pH, no robust calibration curve could be obtained.

## 3. Discussion

In this work, pH was shown to strongly influence *T*_2_ relaxation time constants of ^13^C-labelled hyperpolarizable biomolecules which are commonly imaged in vivo.

[1-^13^C]acetate at 14.1 T showed a minimum of *T*_2_ around pH values close to its p*K*_a_ at 14.1 T. This can be explained by the fast proton exchange of the carboxyl proton in the vicinity of this pH [41,42]. The slope of the titration curve close to this minimum *T*_2_ was slightly flattened towards basic pH values. At pH values close to the p*K*_a_ = 6.81 of water at 37 °C [36], fast proton exchange at the carboxyl group of acetate with non-dissociated water based on the Grotthuss mechanism might take place. Furthermore, the Grotthuss mechanism, describing the exchange of protons between H_2_O molecules, might be most efficient at the mentioned pH, with the exchange rate having spectral density at the Larmor frequency of [1-^13^C]acetate. These effects likely shortened the *T*_2_ values of acetate in the pH range similar to the water p*K*_a_. The higher absolute *T*_2_ values in the basic compared to the acidic regime may be explained by the deprotonated state having one less proton to mediate relaxation via dipole–dipole-interactions. However, acetate ions might still act as weak bases, consequently exhibiting proton exchange with water up to pH 10. Beyond this pH, the acetate ions cannot deprotonate water molecules and *T*_2_ reaches a plateau. At 7 T, there is a weaker sensitivity to pH changes at neutral to slightly basic pH (pH 7–9) compared to 14.1 T, which might be explained by proton exchange with water of the acetate carboxyl group having more spectral density at this Larmor frequency. However, *T*_2_ values at 7 T are generally higher compared to 14.1 T, most likely caused by decreased chemical shift anisotropy effects at lower field. Temperature appears to only have a minor effect on *T*_2_. This, together with *T*_2_ being similar to *T*_1_ [35], indicates that the correlation time of acetate is relatively short due to relatively fast molecular tumbling. This tumbling rate further increases towards higher temperatures, as indicated by the slight increase in *T*_2_ values for increasing temperatures. Salt ions also appear to only slightly shorten *T*_2_, because an increased concentration of the diamagnetic Na^+^-ions induced additional dipolar relaxation mechanisms [43,44]. In contrast, diluting the [1^−13^C]acetate solution to a concentration lower than 250 mM appears to alter *T*_2_ similarly to variations induced by pH, because here the mean distance between hydrated acetate ions in solution becomes too large to still allow hydrogen bond-mediated interactions that potentially enhance *T*_2_ relaxation [45].

Low *T*_2_ values of [1-^13^C]pyruvate at strongly basic pH values are most likely explained by base-mediated keto-enol tautomerism [46], where rapid conformational exchanges and subsequent reactions to para-pyruvate enhance relaxation. At neutral pH values, proton exchange at the carboxyl group might be enhanced by rapid hydroxy–hydronium exchange reactions of water, similar to the observations for [1-^13^C]acetate. This assumption is supported by the observation of decreased *T*_2_ values in the same pH range with added buffer, where rapid proton exchange with buffer molecules might occur, thereby potentially inducing additional relaxation via proton exchange between buffer molecules and [1-^13^C]pyruvate molecules [47]. At strongly acidic pH values, close to pyruvate’s p*K*_a_, fast proton exchange and hydration of the molecule to pyruvate-hydrate further shorten *T*_2_ [48]. The difference in magnitude between the basic global maximum and the local maximum in the weakly acidic regime can be explained by the increased proton concentration at acidic pH, which enhances dipolar relaxation pathways. Regarding the local minimum of the *T*_2_ relaxation time constant at pH 6, proton exchange with water might lead to the observed reduction in *T*_2_. However, this does not fully explain the minimum being at a more acidic pH than the neutral point of water. Further theory models and simulations beyond the scope of this work might be helpful to explain this observation. Regarding [1-^13^C]pyruvate concentration and salt ion concentration, the observations for [1-^13^C]pyruvate are similar to [1-^13^C]acetate and can be explained similarly. As for the temperature dependence of *T*_2_, [1-^13^C]pyruvate shows an exponential decrease with higher temperature, in contrast to [1-^13^C]acetate. Normally, the temperature-dependent molecular tumbling and the related correlation time would lead to an increase in *T*_2_. However, the decrease might rather be explained by the hydration process of [1-^13^C]pyruvate in the pH regime chosen for temperature dependence measurements of *T*_2_. Here, an increased temperature potentially increases the hydration rate of pyruvate [49], and the rapid exchange of water molecules between the pyruvate and the bulk water pool might lead to a reduced *T*_2_ relaxation time, to a sufficient degree to exceed the increase in *T*_2_ due to faster tumbling.

The global *T*_2_ maximum of [1-^13^C]lactate at strongly acidic pH values can be explained by protonated lactic acid molecules forming dimers, rendering the carboxyl proton inaccessible for exchange reactions [50]. At neutral pH, hydrogen bonding of the deprotonated carboxyl group of [1-^13^C]lactate to the intramolecular hydroxyl group and surrounding water molecules might be most efficient [51], explaining the global minimum of *T*_2_. Towards more alkaline pH values, this effect seems to decrease, together with reduced proton concentration, leading to an increase in *T*_2_. At even stronger basic pH regimes, the formation of lactate–metal complexes with added Na^+^-ions and deprotonation of the hydroxy group (p*K*_a_ = 15.8) start to decrease *T*_2_ [52,53].

For [1-^13^C]alanine, pH milieus close to the p*K*_a_ of an amine or carboxyl group of this molecule allow fast proton exchange, which appears to be an effective *T*_2_ relaxation pathway. Furthermore, the overall decrease in *T*_2_ from basic to acidic pH might be explained by susceptibility to the surrounding hydronium ion concentration. In addition, the existence of the amphoteric ion across a wide range of pH values causes the charge distribution in the molecule to be potentially perturbed by dipolar interactions with the charged hydronium ions, thereby promoting *T*_2_ relaxation.

For ^13^C-urea, inability to exchange protons with its environment result in *T*_2_ values being almost inert to variations in pH. The low absolute value of *T_2_*, compared to other investigated ^13^C-labelled compounds, can be explained by the strong quadrupolar relaxation induced by ^14^N nuclei.

For the pH range where [1,4-^13^C_2_]fumarate is dissolvable in water, shortened *T*_2_ values are seen at pH values near the p*K*_a_ of its carboxyl groups, where fast proton exchange at both carboxyl groups occurs, as well as potential influence of proton exchange with water around pH 7. For alkaline pH values, *T*_2_ reaches its maximum as the molecule exhibits as symmetric double deprotonation with limited ability for the formation of hydrogen bonds due to the high concentration of surrounding hydroxyl ions.

Together, the findings in this work are consistent with fast proton exchange being an effective *T*_2_ relaxation pathway, because measured *T*_2_ values decreased strongly in these regimes for proton-exchanging molecules. In this context, buffers contribute to proton exchange processes [47] because the addition of TRIS-buffer or measurements in whole blood, which contains phosphate- and bicarbonate-buffers, were shown to exhibit decreased *T*_2_ compared to unbuffered aqueous solutions. Notably, for all of the proton-exchanging molecules investigated here, their *T*_2_ values were below 50% of their pH-dependent maxima in the vicinity of the p*K*_a_ of water, which has its neutral point at pH 6.81–7.15 [36], for 16–37 °C. Instead, global and local *T*_2_ maxima for all titration curves, excluding ^13^C-urea, were located at moderate to strongly acidic or basic pH. This suggests that water might also act as a proton exchange enhancing moiety, or that it has an increased ability to form hydrogen bonds with the investigated biomolecules [48,54,55] in pH regimes close to its neutral point, which enhances relaxation.

Apart from this strong pH dependence of *T*_2_ relaxation time constants, temperature, concentration and salt ion concentration dependence were investigated to determine their influence on *T*_2_ under conditions close to an in vivo setting. For NaCl concentrations in vivo, ranging at most from 120 mM to 180 mM and normally at 150 mM, a rather low variation of *T*_2_ was observed for [1-^13^C]pyruvate and [1-^13^C]acetate. Temperature variations between 35 °C and 39 °C also appeared to have a limited effect on the *T*_2_ of [1-^13^C]pyruvate and [1-^13^C]acetate. However, concentration dependence of *T*_2_ had a strong effect on in vivo *T*_2_ measurements, where local accumulation and dilution of the tracer, governed by perfusion, might dominate observed variations in *T*_2_. Therefore, in addition to pH, concentration variations might be of crucial influence when mapping *T*_2_ in vivo [15].

*T*_2_ relaxation time constants of ^13^C-labelled biomolecules have already been measured at various field strengths from 3 T to 9.4 T [16,25], within different solvents in vitro [15], and in vivo in several organs [15,16,18,19,25,28]. Previously reported *T*_2_ values of ^13^C-urea [15], [1-^13^C]pyruvate [25] and [1-^13^C]acetate [26] show reasonable agreement with the results from this study, when taking field strength differences and uncertainties in pH and temperature into account. However, in the literature, *T*_2_ measurements in vivo show considerably shortened relaxation times, typically not exceeding a few seconds for most compounds [19,25,27], except for [^13^C, ^15^N_2_]urea (*T*_2_ in vivo up to 11 s [15]). Additionally, multi-compartment relaxation behavior is observed, requiring multi-exponential fitting [15,17,19,20].

While reduced *T*_2_ in vivo compared to in vitro results might be attributed to protein content, oxygenation of hemoglobin [16] and metabolic conversion [18], the presence of multiple *T*_2_ compartments within tumor tissue [20] might also be related to sub-resolution heterogeneity in pH. Interestingly, comparison of measurements between healthy and tumor tissue showed prolonged relaxation times in the latter for [1-^13^C]pyruvate, [1-^13^C]lactate and [1-^13^C]alanine [18,28], while shorter *T*_2_ values were observed in diseased kidneys compared to healthy ones for [^13^C,^15^N_2_]urea [17]. For ^13^C-urea, this was attributed to alterations in tissue oxygenation. For tumor tissue, changes in pH might also contribute to these observed differences. For acidic cancer types, this should be further considered when using spin echo-based sequences, such as bSSFP or RARE, for the imaging of hyperpolarized [1-^13^C]pyruvate, [1-^13^C]lactate and [1-^13^C]alanine, because pH-based prolongation or shortening of the transverse magnetization signal decay might introduce bias to the quantification of metabolic conversion. Simultaneous mapping of pH [6] using hyperpolarized ^13^C-bicarbonate as a metabolic product of [1-^13^C]pyruvate or pH mapping by an additional injection of hyperpolarized [1,5-^13^C_2_]zymonic acid [12] or iopamidol together with chemical exchange saturation transfer (CEST) MRI [56] might be a helpful tool for the estimation of such influences.

Despite these potentially strong effects of pH on existing hyperpolarized ^13^C-MRI imaging quantification approaches, direct application of *T*_2_ mapping of the investigated compound for pH mapping in vivo appears challenging because of the strong influence of concentration on *T*_2_ relaxation.

In this work, slight deviations between *T*_2_ values being measured by a non-localized CPMG train compared to values derived from *T*_2_ mapping could be observed. Here, potentially larger *B*_1_ inhomogeneities and worse shimming for the 3-phantom-geometry might explain these observations. In blood, additional relaxation mechanisms, e.g., caused by paramagnetic deoxyhemoglobin, might have a stronger effect on *T*_2_ than pH. In addition, for injected ^13^C-labelled compounds, large amounts might flow into and out of the imaged volume and contribute to the measured signal, such that refocusing of the magnetization after a certain echo time, and therefore reliable measurement, might be difficult [18]. Nevertheless, for aqueous solutions of known composition, such as quality control in human hyperpolarized ^13^C imaging studies [3], pH control might be realized directly inside the magnet bore via NMR-based *T*_2_ measurements, thereby potentially speeding up the preparation time.

To further investigate the role of water for proton exchange-mediated relaxation processes, titration curves in different solvents or in solvents with varying water content are required. In order to quantify the potential influence of the pH dependence of *T*_2_ for measurements of hyperpolarized ^13^C-labelled metabolites in vivo, tissue pH-targeting therapies using acetazolamide or bicarbonate can be used together with hyperpolarized ^13^C imaging to examine changes in metabolite quantification due to pH alterations.

## 4. Methods

### 4.1. Chemical Compounds

[1-^13^C]Na-acetate, [1-^13^C]alanine and [1,4-^13^C_2_]fumaric acid were obtained from Cambridge Isotope Laboratories Inc. (Tewksbury, Massachusetts, USA). [1-^13^C]pyruvic acid, [1-^13^C]Na-lactate solution (45–55% w/w) and ^13^C-urea were obtained from Sigma-Aldrich (St. Louis, Missouri, USA) and used without further purification.

### 4.2. Sample Preparation

Compounds were dissolved in double distilled (dd) H_2_O (Millipore Milli-Q, Merck, Darmstadt, Germany) for experiments of the non-hyperpolarized state at 1 T and 7 T, unless otherwise stated, and in 10% D_2_O and 90% dd H_2_O for measurements at 14.1 T. For solvents used in measurements with hyperpolarized compounds, refer to Section 4.4., “*Hyperpolarization*”. For each titration curve, stock solutions of 10 mL were prepared, titrated, and measured, and samples put back into the stock after each measurement. If applicable, solutions were buffered with tris(hydroxymethyl)aminomethane (TRIS) (Sigma Aldrich, St. Louis, MI, USA). pH titrations were performed with 10 M KOH solution, different concentrations of NaOH solutions, ranging from 33 mM to 10 M, and different concentrations of HCl solutions, ranging from 33 mM to 12 M to minimize added acid or base volume and to keep changes in ^13^C-compound concentration less than 5%. Titration protocols for measurements at 14.1 T can be found elsewhere [35], and titration protocols for measurements at 7 T are listed in the Appendix A. Titrations with salt were performed by adding weighted amounts of 99.5% NaCl powder (Sigma Aldrich, St. Louis, MI, USA). Measurements in human whole blood were performed on blood samples drawn from C.H. and M.G., which were collected in tubes containing ethylenediaminetetraacetic acid (EDTA) as an anticoagulant.

### 4.3. pH and Temperature Measurement

The pH of NMR samples was measured after spectroscopy measurements using a pH-Combination Electrode N 6000 A and a ProLab 4000 multiparameter benchtop meter (SI analytics, Mainz, Germany). For measurements at 1 T and 14.1 T, the temperature of 600 µL samples was adjusted and maintained automatically with an accuracy of 0.1 °C using the built-in temperature controller of the spectrometer. For measurements at 7 T, the temperature of 2 mL samples was manually adjusted and maintained with an accuracy of 0.5 °C by blowing warm air through the bore using a Pet Dryer Model B-8 (XPower, San Gabriel, California, USA), while temperature was monitored using an MR-compatible temperature monitoring system Model 1030 (SA Instruments Inc., Stony Brook, New York, NY, USA).

### 4.4. Hyperpolarization

[1-^13^C]pyruvate was mixed with 16 mM Ox063 trityl radical (GE Healthcare, Chicago, IL, USA) and 1 mM DOTAREM (Guerbet, Villepinte, France). Amounts of 23.3 ± 3.0 mg of this mixture were polarized for at least 40 min at 1.2 K and 3.35 T using a HyperSense^®^ DNP Polarizer (Oxford Instruments, Abingdon, UK) by microwave irradiation at 94.172 GHz and 100 mW power. Dissolution was performed using 3.28 ± 0.60 mL dd H_2_O containing either 80 mM TRIS (Sigma Aldrich, St. Louis, MI, USA), 80 mM NaOH, 0.1 g/L EDTA or 150 mM Universal buffer (0.1 M disodium phosphate, 0.5 M citric acid), 80 mM NaOH, 0.1 g/L EDTA. Solvents were preheated to 180 °C, resulting in initial solution pH 6.8–7.8 and pH 6.0–7.8, respectively, after dissolution. Further pH data points of titration curves were obtained by rapid titration with NaOH and HCl directly after dissolution.

### 4.5. C NMR Spectroscopy

#### 4.5.1. Measurements at 1 T

For measurements of hyperpolarized samples, a 43 MHz Spinsolve carbon benchtop spectrometer (Magritek, Aachen, Germany; Wellington, New Zealand) was pre-shimmed using a tube containing 10% D_2_O and 90% dd H_2_O. Trains of 4096 spin echoes were acquired using a Carr-Purcell-Meiboom-Gill (CPMG) sequence [13,57] with a short 10 ms echo spacing (to minimize influence of diffusion effects; see Appendix B), receiver bandwidth 200 kHz, 32 spectral points, excitation flip angle 90°, and refocusing flip angle 180°.

#### 4.5.2. Measurements at 7 T

All acquisitions were performed on a 7 T small animal preclinical scanner (Agilent Discovery MR901 magnet and gradient system, Bruker AVANCE III HD electronics) and a dual-tuned ^1^H/^13^C volume resonator (inner diameter 31 mm, RAPID Biomedical). Samples were localized and centered using a *T*_1_-weighted ^1^H spoiled gradient recalled echo (FLASH) imaging sequence and shimmed with ^1^H signal. The reference frequency for each ^13^C sample was determined by manually adjusting while observing the spectrum with a non-spatially selective pulse-and-acquire free induction decay (FID) spectroscopy sequence. The reference power for ^13^C was calibrated using similar ^13^C-FID acquisitions after non-spatially selective excitations with a 1 ms block pulse of varying radiofrequency (RF) power and fitting the complex signal vs. excitation power curves for the 180° phase inversion point, using an in-house-written script in MATLAB (The Mathworks Inc., Natick, MA, USA). After tube temperature was stable, trains of 8192 echoes were acquired using a CPMG sequence with 10 ms echo spacing, receiver bandwidth 5 kHz, 32 spectral points, excitation flip angle 90°, refocusing flip angle 180°, repetition time 300 s, and one average for concentrations of 250 mM or more, or 25 averages for 50 mM-concentrated tubes. For ^13^C imaging, three tubes of 250 mM [1-^13^C]acetate containing solution were prepared and pH was adjusted to three different values in the vicinity of physiological pH using NaOH and HCl. For *T*_2_ mapping, tubes were grouped, and localization and pre-adjustments were performed as described above. Multiple echo images for *T*_2_ maps were acquired on a single axial slice using a single-shot rapid acquisition relaxation enhancement (RARE) sequence with echo spacing 10 ms, 16,384 spin echoes per excitation to repeatedly measure the 16 × 16 acquisition matrix at 1024 effective echo times spaced 160 ms apart, repetition time 300 s, acquisition matrix 16 × 16, field-of-view 32 × 32 mm^2^, slice thickness 16 mm, nominal RARE-factor 16, 144 averages, and receiver bandwidth 3.2 kHz.

#### 4.5.3. Measurements at 14.1 T

All acquisitions were performed on a Bruker Avance III 600 MHz NMR spectrometer (Bruker, Billerica, Massachusetts, USA). After temperature equilibration, for each *T*_2_ measurement, ten acquisitions with a varying number of 1–5000 spin echoes were performed using a CPMG sequence, and only the last echo of each echo train was recorded. Sequences used echo spacing 10 ms, receiver bandwidth 3 kHz, 65,536 spectral points, excitation flip angle 90°, and refocusing flip angle 180°.

### 4.6. Data Processing and Analysis

Data from acquisitions at 1 T and 14.1 T were analyzed using MNova (Mestrelab, Santiago de Compostela, Spain). Data from acquisitions at 7 T were analyzed using in-house-written MATLAB (The Mathworks Inc., Natick, MA, USA) scripts. For *T*_2_ calculation, peaks from spectra of single echo acquisitions at 1 T and 14.1 T were integrated, whereas for data from acquisitions at 7 T, only peak maxima were taken due to low spectral resolution. For the resulting signal decay curves, least-squares fitting with a mono-exponential decay curve including a constant offset was performed. For RARE imaging, echo train decay curves were least-squares fitted voxel-wise analogously to non-localized measurements. For the generation of pH maps, a linear relationship between *T*_2_ and pH was assumed, and the conversion formula was fitted using the relevant section of the pH titration curves. All reported *T*_2_ values come with a statistical uncertainty of 0.1 s, as was derived from iterative acquisition on the same phantom which is described in detail in the Appendix A.

## 5. Conclusions

In this work, the pH dependence of the apparent transverse relaxation time constant *T*_2_ of small hyperpolarized ^13^C-labelled biomolecules was investigated. Titration series revealed strong changes in *T*_2_ with varying pH, typically resulting in minima for pH regimes close to the p*K*_a_ of the proton exchanging compounds. Added buffers and water as proton exchange catalysts, or the latter forming hydrogen bonds, also considerably shortened *T*_2_ at their respective p*K*_a_ values. Variations in field strength, temperature, or ion concentration only had a limited effect on *T*_2_. The observed sensitivity of *T*_2_ to pH, especially for [1-^13^C]acetate and [1-^13^C]pyruvate, can be exploited to generate pH maps from *T*_2_ mapping via RARE acquisitions in aqueous solutions in vitro and shows that pH variations, as might be the case in vivo, can introduce a bias to signal quantification in *T*_2_-sensitive imaging acquisitions.

## Figures and Tables

**Figure 1 pharmaceuticals-14-00327-f001:**
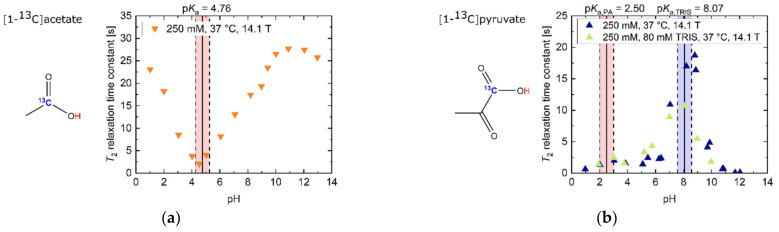
Titration curves of apparent *T*_2_ relaxation time constants for several ^13^C-labelled, thermally polarized biomolecules in H_2_O at 37 °C and 14.1 T. The range of ±0.5 pH steps around the p*K*_a_ values of all molecules is indicated in red background color. For [1-^13^C]pyruvate, the range of ±0.5 pH steps around the p*K*_a_ values of TRIS buffer is indicated in blue background color. (**a**) [1-^13^C]acetate exhibits long *T*_2_ values away from the p*K*_a_ = 4.76. Here, the protonated state, occurring at acidic pH values, has a slightly longer *T*_2_ than the deprotonated state, which exists at basic pH values. (**b**) *T*_2_ of [1-^13^C]pyruvate at extreme basic pH is decreased by one order of magnitude relative to slightly basic pH values around pH 9. Addition of 80 mM TRIS buffer reduces *T*_2_ values in the vicinity of the buffer p*K*_a_ (pH 7–9). (**c**) [1-^13^C]lactate shows its longest *T*_2_ relaxation time constants for its protonated state below pH 3, as well as a local maximum around pH 11, while exhibiting shorter *T*_2_ values at neutral and strongly basic pH. (**d**) [1-^13^C]alanine shows a decrease in *T*_2_ from basic towards acidic pH, with additionally increased relaxation at its p*K*_a_ values. (**e**) ^13^C-urea shows pH-independent, but very low, *T*_2_ values compared to the other investigated compounds. (**f**) [1,4-^13^C_2_]fumarate’s *T*_2_ shows a four-fold increase in the double deprotonated state (basic pH) compared to pH milieus close to the p*K*_a_ values of both carboxyl groups (pH ~4).

**Figure 2 pharmaceuticals-14-00327-f002:**
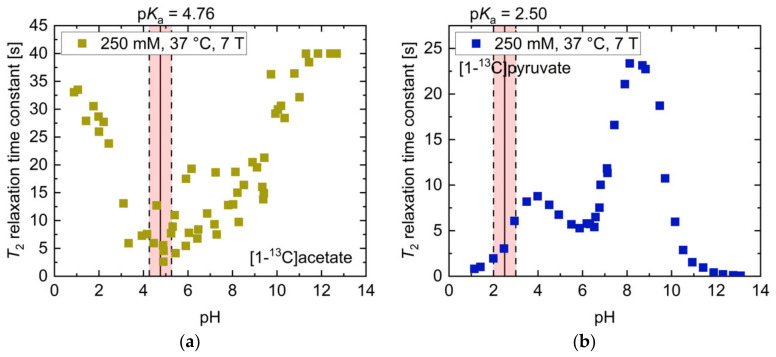
pH titration curves of 250 mM thermally polarized [1-^13^C]acetate and 250 mM [1-^13^C]pyruvate at 37 °C and 7 T. (**a**) The pH dependence of [1-^13^C]acetate qualitatively resembles that observed at 14.1 T (Figure 1a), but shows higher absolute *T*_2_ relaxation times and a linear increase from pH close to its p*K*_a_ to strongly basic pH values. (**b**) The titration curve of [1-^13^C]pyruvate exhibits two maxima at slightly basic and moderately acidic pH and a local minimum at pH 6, together with drastically shortened *T*_2_ values in the strongly acidic and basic regimes. The range of ±0.5 pH steps around the p*K*_a_ values of the molecules is indicated in red.

**Figure 3 pharmaceuticals-14-00327-f003:**
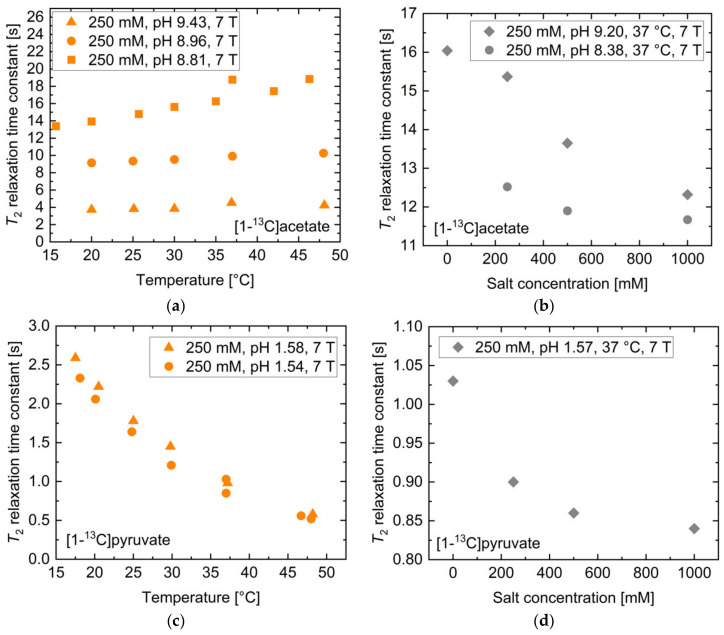
Influence of temperature and salt concentration on *T*_2_ of thermally polarized [1-^13^C] acetate and [1-^13^C]pyruvate. While [1-^13^C]acetate shows a linear increase in *T*_2_ with temperature (**a**), *T*_2_ of [1-^13^C]pyruvate decreases exponentially when increasing solution temperature (**c**). Salt ion concentration leads to minor reductions in *T*_2_ for both molecules (**b**,**d**).

**Figure 4 pharmaceuticals-14-00327-f004:**
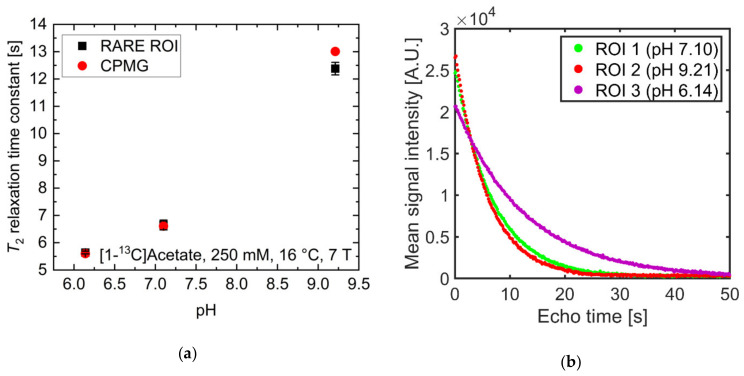
(**a**) Comparison of *T*_2_ values of [1-^13^C]acetate measured with a Carr-Purcell-Meiboom-Gill (CPMG) sequence separately for each tube (red) and derived from averaging *T*_2_ values within regions of interest analysis of *T*_2_ maps generated from rapid acquisition with relaxation enhancement (RARE) imaging data (black). Mean relaxation times show good agreement between acquisition modalities, despite RARE-derived *T_2_* values for each tube being slightly shorter than the corresponding CPMG-derived values. (**b**) Echo trains in tubes of different pH values. Signal curves were derived from echo images and averaged across regions of interest (green, red, or magenta). They show good preservation of transversal magnetization in later echoes. (**c**) *T*_2_ map generated from voxel-wise mono-exponential fitting of echo trains, showing homogeneity within and high contrast between tubes. (**d**) pH map generated using an estimated linear relationship between *T*_2_ and pH, derived from the titration curve in Figure 2a. In this manner, the pH of the tubes can be estimated with 0.1 pH unit intra-tube voxel-by-voxel variation and with an absolute accuracy of 1.6 pH units.

**Table 1 pharmaceuticals-14-00327-t001:** Minimum and maximum *T*_2_ values of 250 mM ^13^C-labelled biomolecules and pH-sensitivity of *T*_2_ in the physiological range pH 6.8–7.8 at 14.1 T.

	*T*_2,max_ [s] (pH)	*T*_2,min_ [s] (pH)	*T*_2_ [s] (pH 7.4)	Ratio *T*_2,max_/*T*_2,min_	Mean Δ*T*_2_ [s/0.1pH] (pH Range 6.8–7.8)
[1-^13^C]acetate	27.7 (10.90)	2.1 (4.52)	14.7	13.2	0.44
[1-^13^C]alanine	10.6 (12.45)	0.6 (5.06)	4.6	19.0	0.09
[1,4-^13^C_2_]fumarate	18.9 (12.91)	3.0 (5.73)	13.9	6.3	0.19
[1-^13^C]lactate	12.6 (1.03)	0.7 (6.99)	1.1	18.9	−0.05 (pH 6.8–7)/ 0.08 (pH 7–7.8)
[1-^13^C]pyruvate	18.7 (8.81)	0.1 (11.69)	13.9	208.1	0.95
^13^C-urea	0.1 (7.79)	0.1 (2.99)	0.1	1.2	<0.001

Note: For *T*_2,max/min_ values, the corresponding pH is indicated in brackets. *T*_2_ values at pH 7.4 were calculated from the interpolation of data points in the close pH range. “Mean Δ*T*_2_” indicates the mean change in *T*_2_ in seconds per 0.1 pH step in the pH range 6.8–7.8, which is of interest for physiological conditions. All *T*_2_ values are rounded to the first digit after the decimal point.

**Table 2 pharmaceuticals-14-00327-t002:** Influence of concentration, salt concentration, temperature, and pH within common physiological and pathological variations on *T*_2_ relaxation time constants of [1-^13^C]acetate at 7 T.

**[1-^13^C]acetate**	***T*_2,ref_ [s]**	***T*_2,lower bound_ [s]**	***T*_2,upper bound_ [s]**	**Rel. Change Lower Bound [%]**	**Rel. Change Upper Bound [%]**
Concentration	20.8	26.1	11.4	25.2	−45.4
Salt concentration	14.8	15.1	14.5	2.0	−1.8
Temperature	11.4	11.3	11.6	−1.6	1.5
pH	11.7	9.7	13.0	−16.9	11.2

Note: Parameter ranges and physiological reference values which are assumed to commonly apply to in vivo experiments in healthy subjects: concentration, reference 10 mM, lower bound 5 mM, upper bound 80 mM; salt concentration, reference 150 mM, lower bound 120 mM, upper bound 180 mM [40]; temperature, reference 37 °C, lower bound 35 °C, upper bound 39 °C; pH, reference 7.40, lower bound 6.8 [37], upper bound 7.8 [38].

**Table 3 pharmaceuticals-14-00327-t003:** Influence of concentration, salt concentration, temperature, and pH within common physiological and pathological variations on *T*_2_ relaxation time constants of [1-^13^C]pyruvate at 7 T.

[1-^13^C]pyruvate.	*T*_2,ref_ [s]	*T*_2,lower bound_ [s]	*T*_2,upper bound_ [s]	Rel. Change Lower Bound [%]	Rel. Change Upper Bound [%]
Concentration	2.1	2.1	1.6	1.9	−23.6
Salt concentration	0.9	1.0	0.9	8.5	−2.1
Temperature	0.9	1.0	0.9	10.6	−9.6
pH	15.4	9.1	19.7	−41.2	27.4

Note: Parameter ranges and physiological reference values which are assumed to commonly apply to in vivo experiments in healthy subjects: concentration, reference 10 mM, lower bound 5 mM, upper bound 80 mM; salt concentration, reference 150 mM, lower bound 120 mM, upper bound 180 mM [40]; temperature, reference 37 °C, lower bound 35 °C, upper bound 39 °C; pH, reference 7.40, lower bound 6.8 [37], upper bound 7.8 [38].

## Data Availability

The data presented in this study are available on request from the corresponding author.

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
