# Peer review of "pH Dependence of T2 for Hyperpolarizable 13C-Labelled Small Molecules Enables Spatially Resolved pH Measurement by Magnetic Resonance Imaging"

_pharmaceuticals, 2021, doi:10.3390/ph14040327_

Round 1
Reviewer 1 Report
The manuscript by Shilling and coworkers presents a systematic study of the T2 dependence on the pH for 13C-labelled small molecules.
Specifically, the authors investigate a series of compounds that are typically used as biomarkers for hyperpolarized imaging.
Most of them show a strong T2 vs. pH dependence. Among those, acetate and pyruvate show the largest pH dependence within the physiological pH limits.
These variations are interpreted qualitatively in terms of proton exchange and the impact that this mechanism has on the relaxation paths.
Finally, the authors nicely show that, in MRI, T2 can be used as an indicator to discriminate between regions with different pH. Considering the target molecule, this proof of concept might have an application in hyperpolarized MRI.
The authors also clearly explain the challenges of such a method, especially in vivo, where the T2 is affected by many more parameters other than the pH.
In my opinion, the manuscript is suited for publication, but the authors should address two issues of the current data presentation.
1) A statement on the errors on T2 measurements is clearly missing, as well as error bars. In this respect, a short discussion concerning the significance of the observed differences in T2 is also needed. Would the data interpretation change regarding, for instance, Figure 2a?
2) It is not clear through the text what is measured under hyperpolarization condition and what at the thermal equilibrium. To my understanding, the only measurements performed on a hyperpolarized compound (pyruvate) are reported in Appendix B, while the rest appears to be done under thermal equilibrium condition. If this is true, then the title is misleading, because there is no hyperpolarization (except for Appendix B) and no hyperpolarized MRI in the manuscript.
Furthermore, how is the radical removed after the hyperpolarization?
Minor) In Figure 3, please use colors or symbols to distinguish the different pH values.
Reviewer 2 Report
The authors thoroughly investigated transversal relaxatiton times of 13C NMR/MRI signal of several simple organic compounds, which could be imaged in vivo by MRI, especially when hyperpolarized. The work is relevant for application of d-DNP, which is one of modern approaches in MRI, enabling mapping of biochemical processes via metabolite distribution. Therefore, the topic of the paper is of interest of scientific groups working in the fields of molecular imaging and imaging techniques.
To my opinion, the paper could be published after some alteration will be made.
General:
Relaxation times are presented with unreasonable accuracy (two decimal digits) – simple look of data spread in the charts (e.g. Fig 1f, pH >8 or 2a, pH 5-11) shows that much larger ESD should be estimated. A high accuracy is maybe given only by a mathematical fit of monoexponential function (if a high number of data was acquired, Fig. 4b), but from only a single experiment...With this respect, a discussion about decrease in T2 of lactate in pH range 6.8-6.9 before further increase is based on a weak experimental evidence (Fig. 1c, p5,l179).
Main objection points to experiment design: a strong effect of 1. probe concentration, 2. ionic strength, 3. pH on T2 was observed. However, these influences are somewhat synergic (as also authors said on p10,l339-343 and p12,l455-458) – so, measuring pH through determination of T2 is almost impossible, if other parameters (concentration, presence of additional compounds/salts) are not exactly known. So, at least two-dimensional experiment should be done, with changing concentration/pH at given (physiologic) ionic strength, or performed e.g. in plasma, similarly as presented in Appendix B, but acquired on the one scanner (constant magnetic field). Such experiment would complement presented results, if it would be easily accessible (but a number of presented results are large, so such additional experiments are not necessary).
Specific topics:
As presence of NaCl have an effect of T2 (compare values in Table 2), drop in T2 found in extreme basic regions of titration curves (Fig. 1a) could be possibly also an effect of increased ionic strength (discussion on p10).
Why only 3 data points were measured in trial to develop pH mapping by MRI? There is lack in data for claim “cannot be perfectly represented with a linear relation” (p9,l312).
p9,l13-14: I am lost... green ROI according to coloured scale corresponds to pH 7.5-7.8 (Fig. 4), according to adjusted pH in the phantom 9.21 (Fig. 4), but authors declare 5.87... The same for other ROIs... What is meant by “absolute accuracy of 1.6 pH units”?... please, better explain this paragraph.
p10,l324-327: Too speculative, if prototropic exchange of carboxylic acid/carboxylate would be significantly faster with hydroxide than with oxonium, and it will matter, an indication of plateau should be seen... Concentration of non-dissociated H2O is much higher than these of H3O+ and OH-, so, influence of ions in near-to-neutral region can be expected to be negligible...
p10,l328-329: According to distribution diagram, concentration of AcO- anion reaches 100% at much lower pH than at about 10, where plateau in Fig. 1a is reached... So, not only “less proton” for dipole-dipole interaction matters in increase of T2, but other effect should take place (viscosity? molecular tumbling?)... Any suggestion?
p13,l525: Please, clarify – what material had temperature of 180.degC? Really solution of buffer?
Formal:
p9,l312: Figure 5d instead of 5d
p11,l384 amine instead of amide
Round 2
Reviewer 2 Report
All objections were adequatelly addressed, I support publication of the manuscript in the present form.